# The Effect of Heat Treatment on the Structural-Phase State and Abrasive Wear Resistance of a Hard-Anodized Layer on Aluminum Alloy 1011

Mykhailo Student [1,*], Iryna Pohrelyuk [1], Juozas Padgurskas [2,*], Volodymyr Hvozdets'kyi [1], Khrystyna Zadorozna [1], Halyna Chumalo [1], Oleksandra Student [1] and Ihor Kovalchuk [1]

[1] Karpenko Physico-Mechanical Institute of the National Academy of Sciences of Ukraine, 79601 Lviv, Ukraine
[2] Department of Mechanical, Energy and Biotechnology Engineering, Vytautas Magnus University, 44248 Kaunas, Lithuania
[*] Correspondence: student.phmi@gmail.com (M.S.); juozas.padgurskas@vdu.lt (J.P.)

**Abstract:** The aim of this study was to evaluate the effect of heat treatment on the phase composition, hardness, and abrasion wear resistance of hard-anodized layers (HAL) on 1011 aluminum alloy. X-ray diffraction analysis revealed the $Al_2O_3 \cdot 3H_2O$ phase in the structure of HAL synthesized for 1 h. While in the heat-treated HAL, aluminum oxide phases of the $\alpha$-$Al_2O_{3(amorphous)}$ and $\gamma$-$Al_2O_{3(amorphous)}$ types were found. Treatment at 400 °C for 1 h increased the HAL microhardness from 400 to 650 HV, and its abrasive wear resistance with fixed abrasive by up to 2.6 times. The ranking of various ways of hardening aluminum alloys relative to the D16 alloy showed that the abrasive wear resistance of heat-treated HAL is 20 times higher. Plasma electrolyte oxidation increased the abrasive wear resistance of the D16 alloy by 70–90 times, and its coating with high-speed oxygen fuel by 75–85 times. However, both methods are complex, energy-consuming, and require fine grinding of parts. Despite the lower wear resistance of HAL, their synthesis is cheaper and does not require the fine-tuning of parts. Moreover, despite the low hardness of HAL at present, hard anodizing is already commercially used to harden engine pistons, clamshell rotators, and pulleys.

**Keywords:** aluminum alloy; hard anodized layers; heat treatment; phase transformations; microhardness; abrasive wear resistance

## 1. Introduction

Aluminum alloys are characterized by low abrasive wear resistance, which prevents their widespread use in industry. To increase their wear resistance, galvanic chromium plating [1], thermal spray coating [2–4], Plasma Electrolyte Oxidation (PEO) [5–9], and Hard Anodized Layer (HAL) formation [10–12] are most often used. The use of carcinogenic and environmentally harmful electrolytes in the implementation of the chromium plating method makes it environmentally hazardous [1,13,14]. PEO is used to form oxide layers on compact aluminum alloys [15–17], to oxidize aluminum coatings deposited in various ways on a steel base [18–22], or on other substrates [23,24]. PEO of aluminum alloys makes it possible to obtain surface layers with high hardness (up to 2000 HV), low friction coefficient, high adhesion to the metal base, and low environmental hazard [25–28]. However, the disadvantages of this method include high energy consumption for its implementation, which makes it impossible to synthesize PEO layers on large-sized elements. The method for the synthesis of HAL is relatively cheap and technologically simple. Therefore, it is widely used in industry. However, due to its significant disadvantages (low hardness ≤500 HV and wear resistance [29–33]), its wider application for surface hardening of various elements is limited.

Hard anodizing is usually carried out at a lower electrolyte temperature, higher current density, and higher voltage than conventional anodizing. Typically, HAL are formed at a





voltage between the electrodes of no more than 80–100 V. The porosity of the outer layers of HAL determines their unsatisfactory protective properties.

Hard anodizing is a highly exothermic process. A local increase in temperature contributes to a local increase in the thickness of the synthesized layer and the inhomogeneity of its microstructure, which worsens its mechanical properties [34,35]. Therefore, it is important that the heat released during the synthesis of HAL be efficiently removed. In particular, it has been shown [36] that the thickness of the HAL depends on the rate of two opposing processes, such as the rate of its synthesis, determined by Faraday's law, and the rate of its dissolution. In general, the rate of HAL synthesis depends on the current density, and its dissolution depends on the chemical composition and temperature of the electrolyte [36,37].

As a rule, hard and wear-resistant HAL are formed at an electrolyte temperature of about 0 °C and a high current density (>2.5 A/dm$^2$) [38–40]. The microhardness of anodized layers synthesized at a higher temperature (11 °C) did not exceed 440 HV [41]. With an increase in the density of the anodizing current from 1 to 4 A/dm$^2$, the diameter of the nano-fibers in the structure of the anodized layer increased from 75.99 $\pm$ 7.7 to 124.59 $\pm$ 6.53 nm, and their number decreased from 6.6 $\pm$ 0.61 to 3.8 $\pm$ 0.48 for a length of $1 \times 10^3$ nm. As a result, the mechanism of tribological wear of HAL during their contact with a ceramic ball changed from the formation of grooves on friction surfaces to micro cutting of surface layers in the contact zone [42].

The results of studies of the mechanical properties of anodized layers of various thicknesse on the surface of 6061-T6 alloy showed that the maximum wear resistance corresponded to the HAL with the greatest thickness [43]. Consequently, the wear resistance of HAL increased when increasing their thickness.

It has been shown that the mechanical properties of hard anodized layers (in particular, hardness) depend on the average current density during their synthesis and the kinetics of oxide dissolution [32]. Thus, even in a relatively short anodization time (30 min), the hardness of the coating increased with an increase in the average current density.

It has been experimentally established that the hardness measured on the cross section of the anodized layers does not correlate with their wear resistance [43]. Based on this, a new approach is proposed to determine the wear resistance of surface-hardened layers based on the results of scratching carried out at a constant load. Analysis of SEM images of wear marks showed signs of an adhesive wear mechanism on the electrochemically polished 1050 aluminum alloy substrate, while these signs corresponded to abrasive wear on the anodized aluminum area.

Comparison of the tribological properties of both the classical hard anodized layers and the one synthesized by micro-arc oxidation showed that the hardness and elastic modulus of the micro-arc oxidation layer are three times higher than those determined for HAL [44]. The wear tests were evaluated by reciprocating an alumina ball over hardened surfaces. Specimens with HAL synthesized on their surfaces did not pass the tests, since their surfaces were completely worn out. The wear rate of the micro-arc oxidation layers was $3.1 \times 10^{-5}$ mm$^3$ N$^{-1}$ m$^{-1}$, which was 22 times lower than for the HAL.

To improve the wear resistance of anodized layers synthesized on the surface of aluminum alloys, various methods are proposed. Among them is the ultrasonic impregnation of the pores of the anodized layers with the C60 lubricating component, which made it possible to reduce the friction coefficient of the HAL from 0.58 to 0.18 [45]. The greatest effect on the hardness of the anodized layers was recorded when hydrogen peroxide was added to the electrolyte or when the electrolyte was bubbling with a mixture of air and ozone. In particular, the effect of bubbling an electrolyte with an ozone–air mixture on the process of anodizing aluminum alloys was studied [46]. Anodizing was carried out at a temperature of (0 $\pm$ 1) °C, at different contents of ozone in a mixture with air (1, 3 and 5 mg/L) and at a concentration of sulfuric acid in the electrolyte from 2.5 to 20%. The microhardness of the synthesized layer increased from 5.2 to 6.8 GPa, and their wear resistance by 1.4–2.3 times as the ozone concentration increased from 1 to 5 mg/L. It was also found that the thickness

of the anodized layer increased by 50% when hydrogen peroxide was added to the base electrolyte (20% aqueous solution of $H_2SO_4$) [33]. In addition, the microhardness of the HALs also increased by 60% (from 400 to 650 HV) with an increase in the concentration of hydrogen peroxide in the electrolyte of the base composition from 0 to 70 g/L.

For more efficient heat removal from the HAL synthesis zone, pulsed anodization is used [47,48]. As a result, pores of a smaller diameter are formed in the anodized layer (compared with anodizing with a direct current), and their number remains unchanged. This contributes to an increase in the wear resistance of HAL and a more uniform distribution of hardness over the thickness of the surface layer. Pulse anodizing at a small coating thickness (18 μm) did not provide the anodized layer with high hardness (442 HV). However, such anodizing made it possible to successfully harden the surface of castings made of ADS14 aluminum alloy and, as a result, to reduce their surface wear when used as elements of a continuously variable transmission [49].

At the same time, despite significant progress in improving the physical and mechanical properties of anodized layers on aluminum alloys, briefly discussed above, there are still unknown ways to increase their hardness and wear resistance to levels corresponding to galvanic chromium plating. If this is successful, hard anodizing could be an environmentally friendly alternative to chrome plating.

Today, the attention of researchers is focused on the development of new and improvement of known methods for the synthesis of HAL. The purpose of all these studies is to increase the growth rate and thickness of anodized layers and to improve their hardness and wear resistance. Heat treatment of HAL is one of the possible ways to improve these properties.

It is known that the higher the temperature of the electrolyte during anodization, the greater the number of water molecules is included in the HAL. At the same time, the microhardness of anodized layers decreases with an increase in the number of water molecules in their composition [33]. Thus, hydrated aluminum oxide $Al_2O_3 \cdot 3H_2O$ (gibbsite) with three water molecules has a layered structure and low microhardness [50]. At the same time, hydrated aluminum oxide $Al_2O_3 \cdot H_2O$ (boehmite) with one water molecule has a dense structure and increased microhardness [51]. Since heat treatment promotes dehydration of the anodized layers, an increase in the hardness of the anodized layers was expected.

The purpose of this work is to establish the effect of heat treatment parameters on the structure, phase composition, microhardness, and abrasive wear resistance of 1011 aluminum alloy with hard-anodized layers formed after different anodizing times.

## 2. Materials and Methods

### 2.1. The Method of Syntheses of Hard-Anodized Layers

The process of hard anodization was carried out in a 20% aqueous solution of $H_2SO_4$ at a current density of 5 A/dm$^2$. The electrolyte temperature during the formation of HAL was maintained at the level of −4–0 °C. The duration of HAL formation was 60, 120, and 180 min. A general view of the installation for the synthesis of hard anodized layers on the surface of aluminum alloy specimens and a schematic diagram of this installation are shown in Figure 1. Specimens for anodizing in the form of plates with a size of 20 × 20 × 5 mm$^3$ were made of technical aluminum 1011 (wt. %: 0.25 Si; 0.40 Fe; 0.05 Cu; 0.05 Mn; 0.05 Mg; 0.05 Ti; the rest is Al), an analogue of which is aluminum alloy AA1050. Before anodizing, the specimens were degreased in an aqueous solution of a mixture (CaO + MgO) and washed in cold and warm water, followed by clarification in an aqueous solution of nitric acid (400 g/L $HNO_3$) for 30 s.

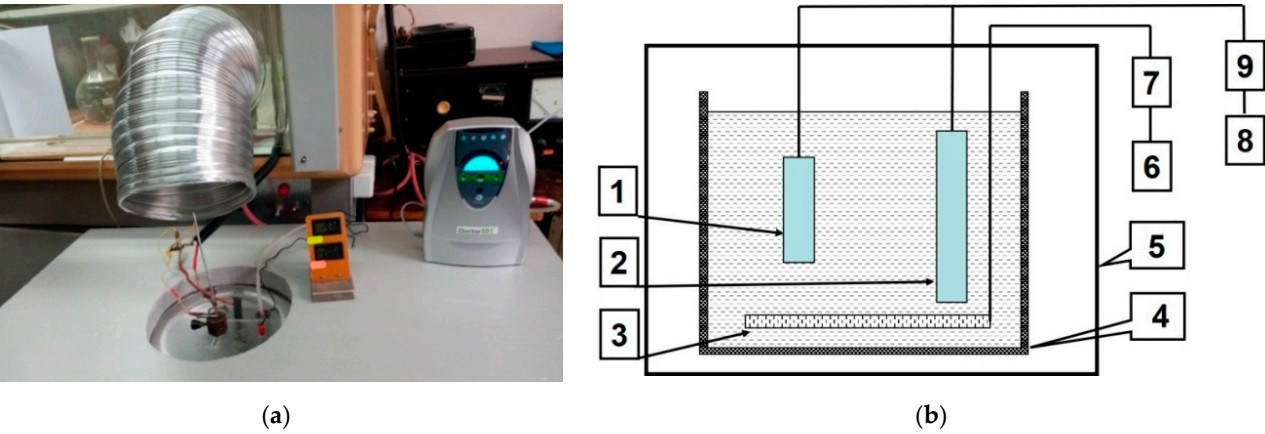

<div align="center">(<b>a</b>)              (<b>b</b>)</div>

**Figure 1.** (**a**) General view of the installation for the synthesis of hard anodized layers on the surface of aluminum alloy specimens; (**b**) and its schematic representation: 1—specimen for anodizing; 2—electrode; 3—bubbler; 4—container with electrolyte (20% aqueous solution of $H_2SO_4$); 5—chamber for stabilizing the temperature regime of the anodizing process in the range of $-4$–0 °C; 6—compressor; 7—block for adjusting the performance of the bubbler and compressor; 8—power supply for maintaining the current density of 5 A/dm$^2$ between the specimen and the electrode; 9—block for adjusting and controlling the power of the power source.

Specimens of aluminum alloy 1011 after the synthesis of HAL on their surfaces were subjected to heat treatment in air in an electric furnace SNOL 30/1100. Its software made it possible to maintain the heating rate (1 °C min) and the selected temperature during the entire treatment period and automatically turn off the power to the furnace at the end of it. The maximum heat treatment temperature was 400 °C; the step of its change was 50 °C. The duration of exposure of the specimens with HAL synthesized on their surface at the appropriate temperature was 1 h after the temperature stabilization in the electric furnace. The temperature inside the furnace was maintained with an accuracy of $\pm 2$ °C. The specimens were heated and cooled together with an electric furnace.

*2.2. Alternative Surface Hardening Methods for Aluminum Alloys Used in the Article to Rank HAL Properties*

Thermal spraying of a coating with a thickness of 500 μm by the method of high-velocity oxygen fuel (HVOF) was carried out on equipment (Diamond Jet Hybrid gun, (Oerlikon Metco Inc., Westbury, NY, USA), in which a mixture of propane and oxygen was used as fuel. Before coating, the surface of specimens made of D16 aluminum alloy (as prototype of AA 2024 alloy) was treated with corundum. The spraying distance was 175 mm, and the speed of powder particles (vanadium carbide + ferrochromium in a ratio of 50/50) with a size of 20–45 μm was 650 m/s.

Arc spray coatings (ASP) 500 μm thick were deposited with an FMI-2 cored wire (Cr6-Al6-B3-Fe the rest) using an FMI metallization device developed at the Karpenko Physico-Mechanical Institute of the National Academy of Sciences of Ukraine (Lviv). The surfaces of the specimens were pre-treated with corundum. Spraying parameters: arc voltage—32 V, current strength—150 A, air jet pressure—0.6 MPa, and spraying distance—130 mm.

Plasma-electrolyte oxidation (PEO) to a depth of 200 μm was carried out on the surface of aluminum alloy D16T in electrolyte (3 g/L KOH + 2 g/L $Na_2SiO_3$) using a pulsed current with a frequency of 50 Hz, at a ratio of the cathode and anode current densities of 15/15 A/dm$^2$. The duration of the PEO process was 60 min.

Two characteristics were used to certify and rank all surface-hardened layers obtained by hard anodizing, HVOF, PEO, and ASC: microhardness HV, measured on a PMT-3 device at a load of 50 g, and abrasive wear resistance by the fixed abrasive method 1/W, determined by weight loss W of specimens obtained on an electronic analytical balance of the KERN ABJ 220 4M type with an accuracy of $2 \times 10^{-4}$ g.

### 2.3. Test Method for Determining the Abrasive Wear Resistance of Hard-Anodized Layers

A device for studying the abrasive wear resistance (with a fixed abrasive) of surface-hardened layers was mounted on a milling machine (as shown on Figure 2a). A schematic diagram of the installation for testing the abrasive wear of specimens with HAL synthesized on their surface is shown in Figure 2b. During the tests, an abrasive disc with a hardness of 2200 HV, made of electrocorundum on a ceramic bond (size of corundum grains was 250–315 μm) with a diameter of 150 mm and a width of 8 mm was used. Its rotation frequency was 2.7 s$^{-1}$, and the load in the zone of linear contact of the disk with the specimen surface was P = (14.7 ± 0.25) N. The friction was 1800 m. To unify the conditions, at the beginning of each test, the cylindrical surface of the abrasive disk was profiled, removing the top layer from it (up to 0.2 mm). For this, a silicon carbide tool was used, the hardness of which was 3500 HV. The test specimen was rigidly fixed on a lever, which transferred the load to the contact zone of the specimen and the abrasive disk. The wear values were estimated from the weight loss $W$ of the specimens after their abrasive wear tests.

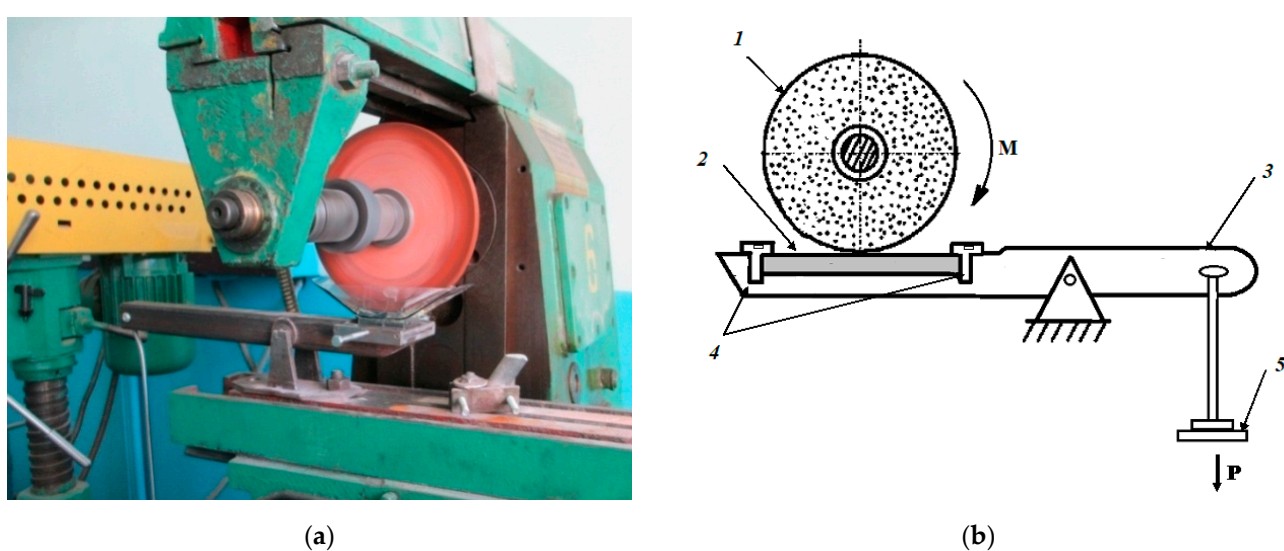

(**a**)            (**b**)

**Figure 2.** (**a**) General view of the installation for abrasive wear testing of specimens with hard-anodized layers synthesized on their surface and (**b**) schematic diagram of this device: 1—abrasive wheel; 2—specimen; 3—lever transmitting force; 4—specimen fixation elements; 5—weights; P—load. Reprinted from Ref. [52].

To rank aluminum alloys with surface-hardened layers, their abrasive wear values were also evaluated in a friction test with a free (non-fixed) abrasive. A scheme of a test setup for assessing the abrasive wear resistance of specimens during tests with non-fixed abrasive is shown in Figure 3. A device for studying abrasive wear resistance (with a non-fixed abrasive) of surface-hardened layers was mounted on a milling machine. The linear speed of the rubber disk with a thickness of 15 mm and a diameter of 50 mm was 25 m/min. The load in the contact zone between the specimen and the disk was 44 N. Dried quartz sand with a particle size of 200–1000 μm was continuously supplied from the container to the contact zone of the rubber disk with the specimen. The duration of testing for each specimen was 30 min.

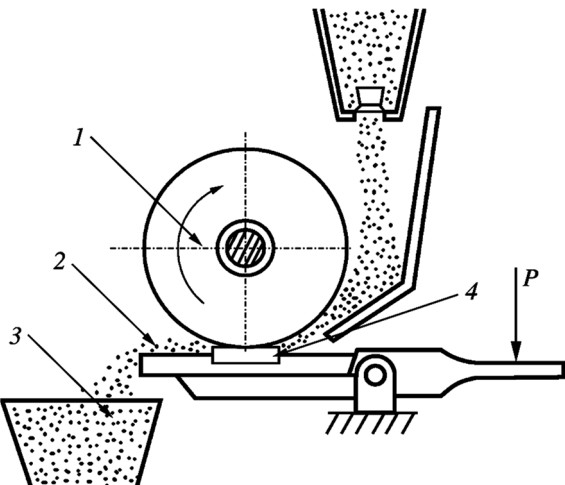

**Figure 3.** Scheme of a test setup for assessing the abrasive wear resistance of specimens during tests with non-fixed abrasive: 1—rubber disk, 2—sand, 3—container for abrasive, and 4—specimen. Reprinted from Ref. [2].

The structure and micro-X-ray spectral analysis of the hard anodized layers were studied using an EVO 40 XVP (Zeiss, Berlin, Germany) electron microscope with an INCA Energy 350 (Oxford Instrument, Abingdon, UK) microanalysis system. A Bruker D8 Discover (Billerica, MA, USA) and DRON-3M (St. Petersburg, Russia) X-ray diffractometers were used for the phase analysis of the synthesized layers.

## 3. Results

### 3.1. Metallographic Analysis of the Structure of Hard Anodized Layers

In surface layers synthesized by the traditional method or by hard anodizing, as a rule, nano- and micron-sized pores are formed [45,53]. Figure 4a,b give an idea of the actual shape and size of nanopores in two sections. Through nanosized pores, the electrolyte penetrates into the barrier layer, etching and stabilizing its thickness at the level of 10–30 nm, as shown in the diagram in Figure 5. Only then can the oxygen ions penetrate the barrier layer and combine with aluminum ions, forming alumina.

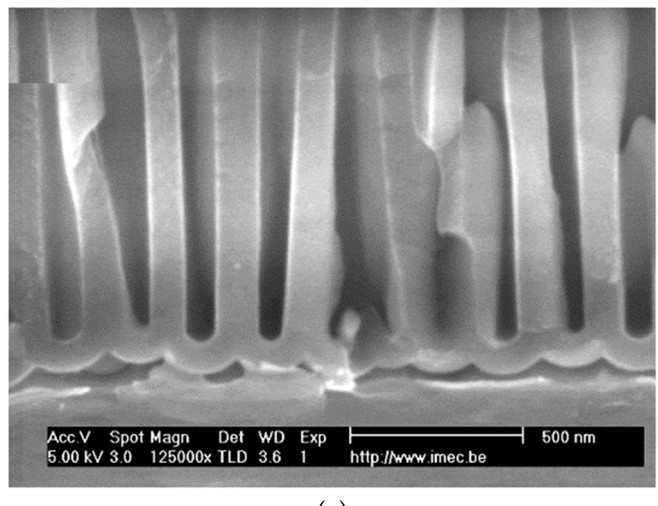

(**a**)

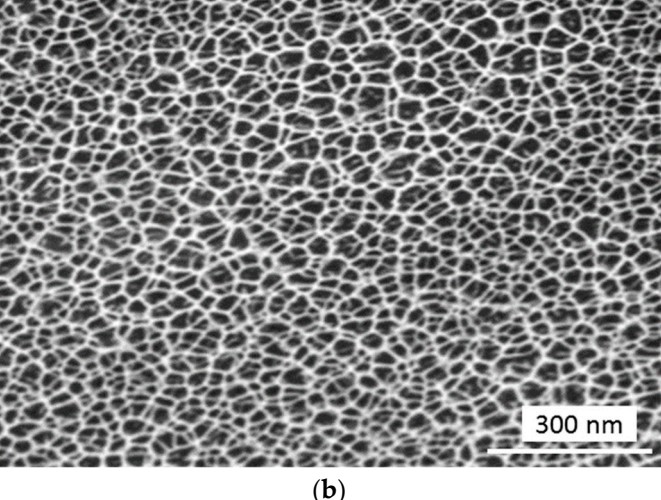

(**b**)

**Figure 4.** Image of real nanosized pores (**a**) in the cross section of the anodized layer; (**b**) their top view. Reprinted from Ref. [53].

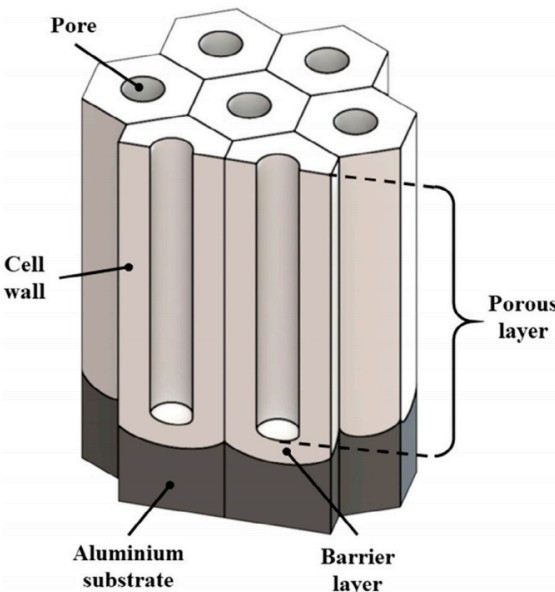

**Figure 5.** Schematic representation of the structure of the anodized layer. Reprinted from Ref. [54].

The appearance of microscale pores in the anodized layer can significantly worsen its corrosion and electrical insulating characteristics. Therefore, their appearance is considered an undesirable sign. Figure 6a shows intermetallics of the Fe(Mn,Cu,Mg)$_3$Al type, which did not dissolve in the solid solution during heat treatment of the 1011 aluminum alloy and remained in the structure of the anodized layer. It is these inclusions that contributed to the formation of microscale pores in the hard anodized layer. Figure 6b shows such pores that were formed in the anodized layer under the influence of electrolyte, which eroded the boundaries of intermetallic inclusions Fe(Mn,Cu,Mg)$_3$Al with the surrounding layer during its synthesis. As a result of such etching, pores commensurate with intermetallic inclusions were formed in their place. The precipitation of copper at the cathode was considered evidence of such etching during anodization of the surface of the aluminum alloy. However, Fe(Mn,Cu,Mg)$_3$Al intermetallides do not usually have time to completely dissolve upon contact with the electrolyte and therefore (as shown in Figure 6c,d) partially remain in the pores of the hard oxidized layer.

Figure 6d illustrates that, as large intermetallic compounds in the base metal approach the interface between the anodized layer and the matrix, the layer thickness opposite to their location is much less than in areas where these inclusions are located deeper from the interface line. Consequently, the intermetallics inside the aluminum base slow down the advance of the anodizing front deep into the aluminum, make it bend, and bypass the intermetallics from all sides. During anodization, these inclusions were partially dissolved in the electrolyte penetrating through nanosized pores in the anodized layer, and due to this, inside it they had a striped morphology, which is a sign of their division into smaller particles.

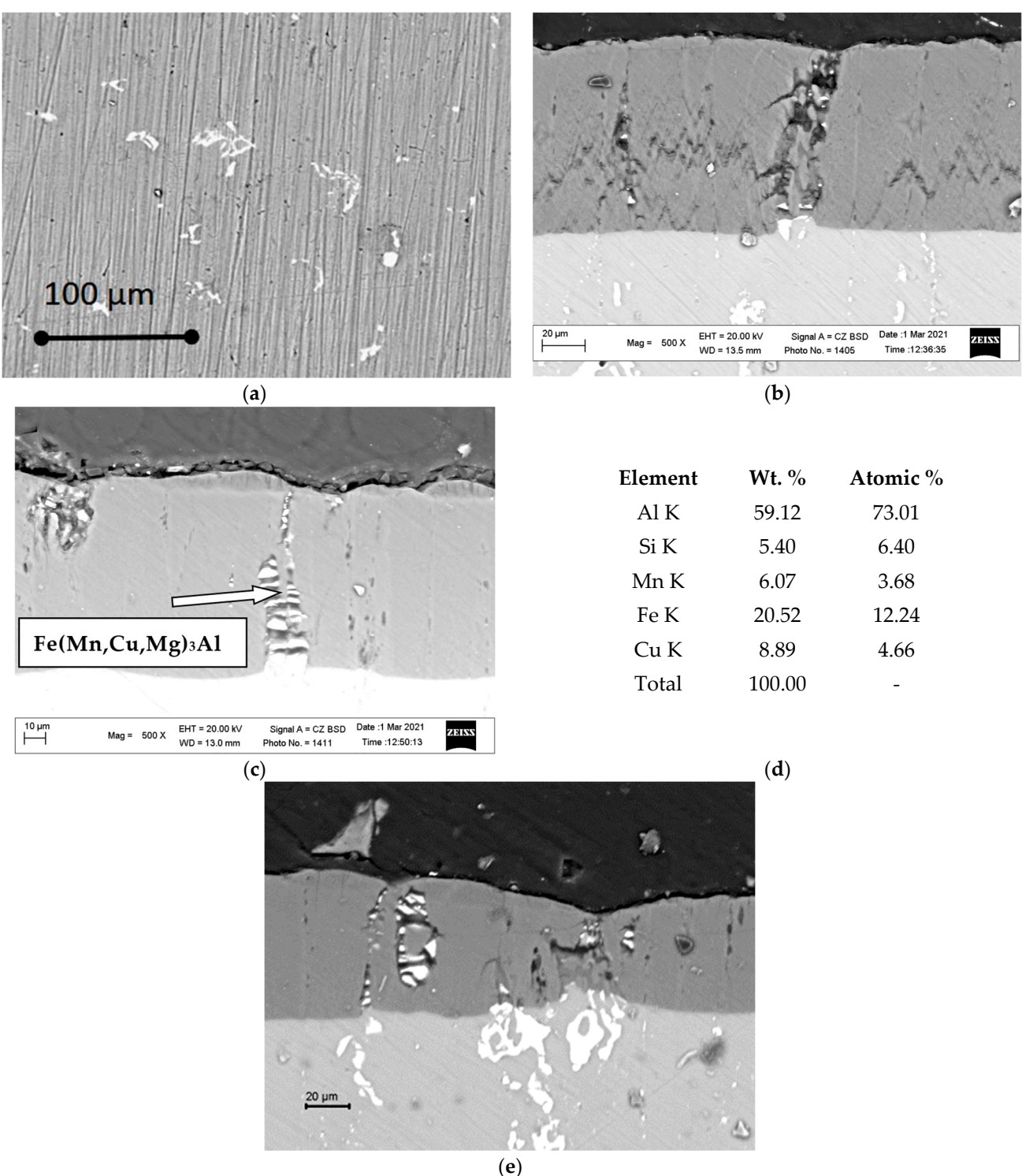

**Figure 6.** (**a**) General view of intermetallic particles on the outer surface of the anodized layer synthesized for 1 h at a temperature of −4–0 °C; (**b**) the formation of micron-sized pores at the site of intermetallic inclusions that appear in the cross section of the anodized layer; (**c**) intermetallic particles and their typical composition; (**d**) the results of X-ray spectral analysis of Fe(Mn, Cu, Mg)3Al intermetallic compounds, indicated by an arrow in the photomicrograph in Figure 6c; (**e**) bending of the anodizing front as it approaches large inclusions in the structure of the base 1011 aluminum alloy.

### 3.2. Phase Analysis of Anodized Layers Synthesized on the Surface of 1011 Aluminum Alloy

Nanosized $Al_2Cu$ inclusions, which are the main hardening factors of aluminum alloys, are isolated in the structure of the anodized layer during artificial or natural aging. The results of X-ray microspectral analysis of the area of the anodized layer on the 1011 alloy, shown in Figure 7, indicates that the oxide phase of the $Al_2O_3$ matrix contains a small amount of magnesium and copper, but a significant amount of sulfur. An increased content of copper was recorded in the lower part of the synthesized layer. This is due to the less intense dissolution of the $Al_2Cu$ intermetallic inclusions in this part of the anodized layer due to the limited access of the electrolyte to them during the anodizing process.

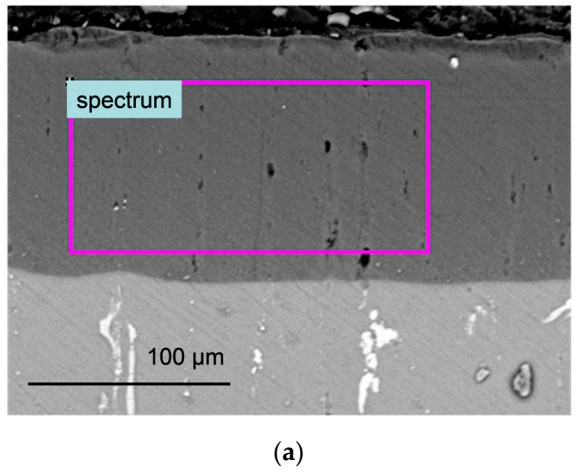

| Element | Weight % |
|---------|----------|
| O | 47.54 |
| Mg | 0.54 |
| Al | 44.43 |
| S | 6.64 |
| Cu | 0.84 |
| Total | 100.00 |

(**a**)  (**b**)

**Figure 7.** (**a**) Micrograph of an anodized layer synthesized on the surface of aluminum alloy 1011 at an electrolyte temperature of −4–0 °C for 1 h with a highlighted area for micro-X-ray spectral analysis and (**b**) the results of the analysis of the content of elements in this zone.

The result of the phase analysis of the anodized layer on the surface of the 1011 alloy, synthesized for 1 h at an electrolyte temperature of −4–0 °C, is shown in Figure 8. Reflexes of aluminum oxide with three water molecules were found on the X-ray pattern. It is believed that the anodized layer containing three water molecules is characterized by the gibbsite—$Al_2O_3 \cdot 3H_2O$ crystallographic lattice [55].

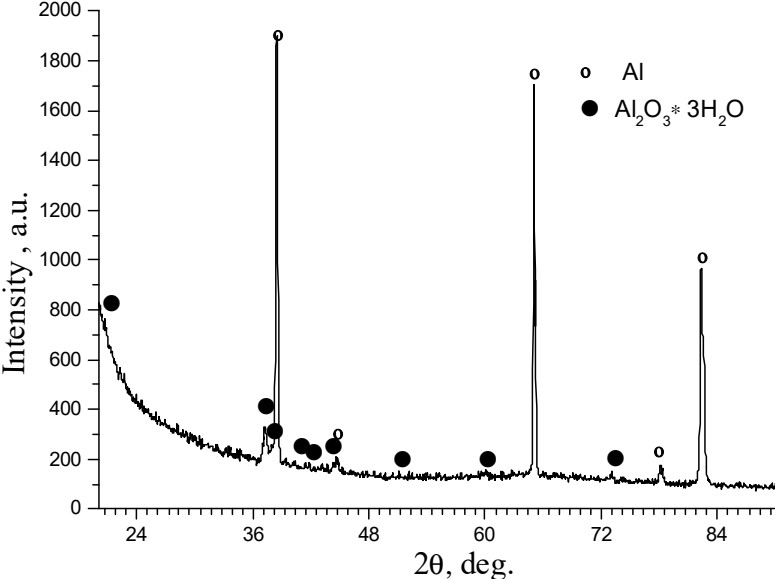

**Figure 8.** Typical phase composition of hard anodized layers synthesized for 1 h at at an electrolyte temperature of −4–0 °C.

To increase the thickness of the anodized layers obtained during the synthesis for 1 h, the duration of the synthesis was increased to 2 and then to 3 h. X-ray phase analysis showed that already after 2 h of synthesis, reflections of aluminum oxide (of the $Al_2O_3 \cdot H_2O$ type) appeared on the X-ray diffraction pattern of the anodized layer shown in Figure 9a, which already contained only one water molecule. It was believed that such dehydration of the hard anodized layer contributed to its compaction, which could have a positive effect on its microhardness.

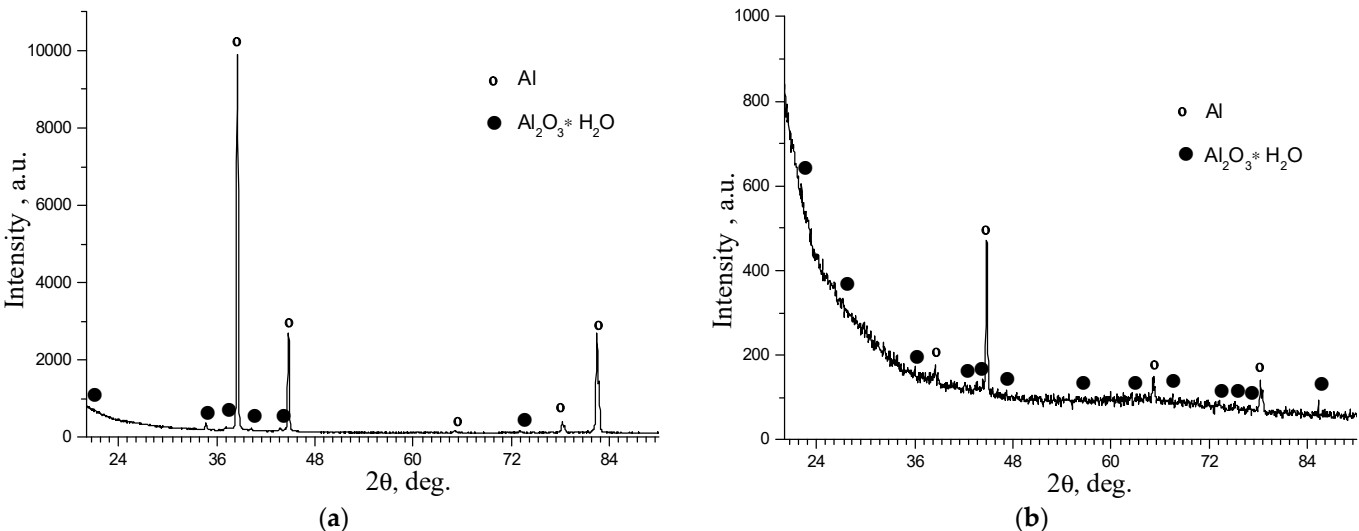

**Figure 9.** Typical phase compositions of hard anodized layers synthesized: (**a**) for 2 h and (**b**) 3 h at an electrolyte temperature of −4–0 °C.

Phase analysis of a hard anodized layer synthesized for 3 h, the results of which are presented in Figure 9b, confirmed the presence of aluminum oxide containing only one water molecule. In this case, the intensity of the reflections corresponding to $Al_2O_3 \cdot H_2O$ was significantly lower than on the specimens after 2 h of synthesis. This was considered a sign of further dehydration of the anodized layer, and, consequently, its further compaction.

It is known that the corrosion resistance of anodized parts is increased by keeping them in boiling water [56]. It is believed that aluminum hydroxide is formed on the walls of the pores during such an exposure. As a result, the porous hard anodized layer becomes more compact. The nanosized pores are easily compacted, which makes it possible to significantly increase the corrosion resistance of an aluminum alloy with a surface layer obtained by hard anodizing. However, the anodized layer with micron-sized pores cannot be compacted sufficiently. This prevents a significant improvement in the corrosion resistance of the anodized layers.

The described results testify to the prospects of using heat treatment of specimens with hard anodized layers to improve their functional properties. As follows from Figure 10a, X-ray phase analysis revealed in the hard anodized layer (synthesized for 2 h followed by heat treatment at 300 °C for 1 h) amorphous phases of aluminum oxides $\alpha\text{-}Al_2O_{3(amorphous)}$ and $\gamma\text{-}Al_2O_{3(amorphous)}$. As shown in Figure 8, with an increase in the duration of synthesis to 3 h (to obtain a thicker anodized layer) followed by heat treatment (at 300 °C for 1 h), the amount of the amorphized phase $\alpha\text{-}Al_2O_{3(amorphous)}$ increased, and the amount of $\gamma\text{-}Al_2O_{3(amorphous)}$, on the contrary, decreased relative to those obtained during the synthesis of the layer for 2 h. The amorphous phase of aluminum oxide $\alpha\text{-}Al_2O_{3(amorphous)}$ is characterized by a higher hardness than $\gamma\text{-}Al_2O_{3(amorphous)}$. This gave reason to expect that heat treatment can increase the hardness of the anodized layers.

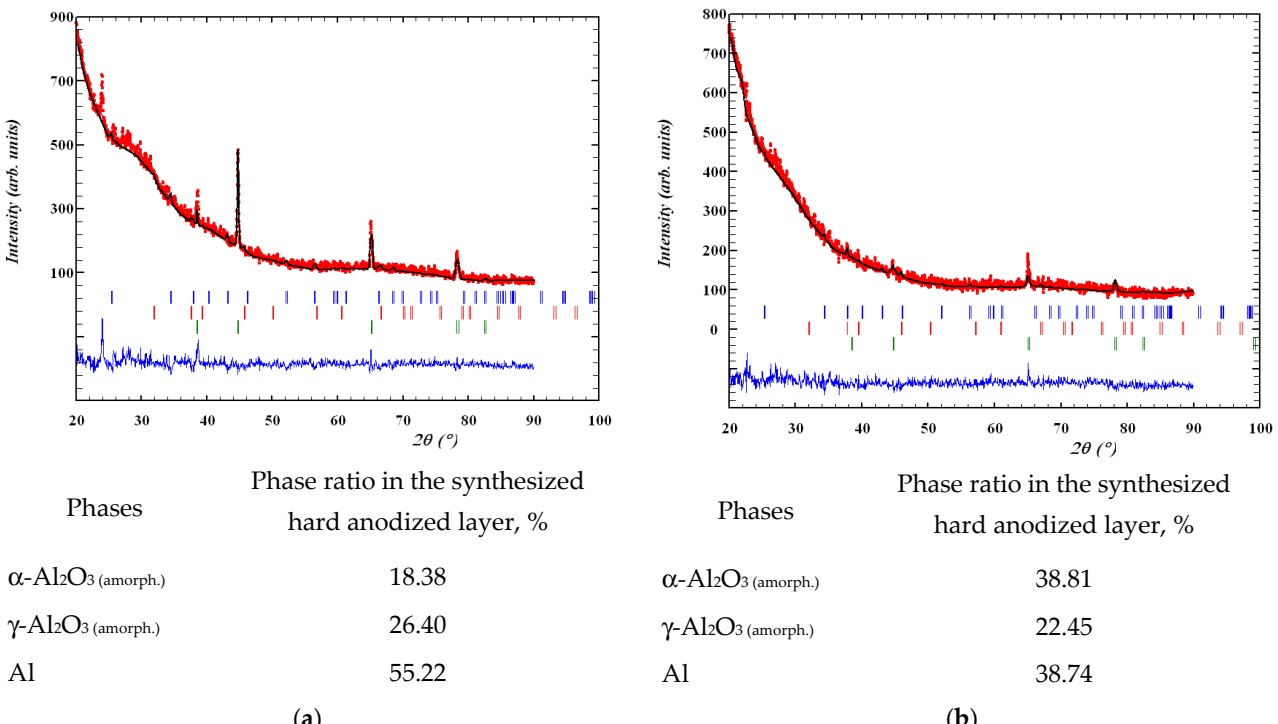

| Phases | Phase ratio in the synthesized hard anodized layer, % | Phases | Phase ratio in the synthesized hard anodized layer, % |
|---|---|---|---|
| α-Al₂O₃ (amorph.) | 18.38 | α-Al₂O₃ (amorph.) | 38.81 |
| γ-Al₂O₃ (amorph.) | 26.40 | γ-Al₂O₃ (amorph.) | 22.45 |
| Al | 55.22 | Al | 38.74 |
| (**a**) | | (**b**) | |

**Figure 10.** X-ray diffraction patterns and phase ratios in hard-anodized layers synthesized (**a**) for 2 and (**b**) 3 h on specimens of aluminum alloy 1011 followed by their heat treatment at 300 °C for 1 h.

### 3.3. The Hardness HV of Hard Anodized Layers on the Surface of Aluminum Alloy 1011

Figure 11a shows that the hardness of the anodized layer synthesized within 1 h did not exceed 400 HV. At the same time, heat treatment of specimens with layers synthesized in the same mode, followed by their exposure for 1 h at a temperature of 100 °C, increased the hardness of the anodized layers to 480 HV. This could be a sign of the beginning of the release of water from the anodized layer. Without the use of heat treatment, this water was inside this layer in a crystal-bound state. With an increase in the heat treatment temperature of specimens with anodized coatings to 300 °C, their microhardness increased to 630 HV. This was explained by the intensification of the process of dehydration of the hard anodized layers. As shown in Figure 11a, at temperatures above 300 °C, the increase of the coating hardness slowed down. As a result, after heat treatment at a temperature of 400 °C, the microhardness of the anodized layers increased only up to 650 HV. The maximum increase in microhardness after such heat treatment somewhat exceeded 60%. But after treatment at 300 °C, this increase almost reached 60%. Therefore, to increase the microhardness of the anodized layers, it makes no sense to increase the temperature of their heat treatment above 300 °C.

From Figure 11b, it is obvious that even without heat treatment, but with a longer synthesis time (2 h), the microhardness of the anodized layer increased to 560 HV. Thus, an increase in the hardness of specimens with an anodized layer can be achieved either by applying heat treatment at a temperature of 300 °C (with a duration of the synthesized layer of 1 h) or by increasing the duration of synthesis to 2 h. However, the microhardness of specimens with an anodized layer synthesized within 2 h can be further increased (up to 660 HV) by additional heat treatment at 300 °C. An increase in the duration of anodization up to 3 h did not reveal a significant increase in the microhardness of the hard anodized layer. Additional heat treatment at 300 °C for specimens with anodized layers (after 3 h of synthesis) also had little effect on their hardness. Therefore, it was concluded that the results shown in Figure 11b prove the inexpediency of using a 3 h synthesis for hard anodization of the surface of a 1011 aluminum alloy.

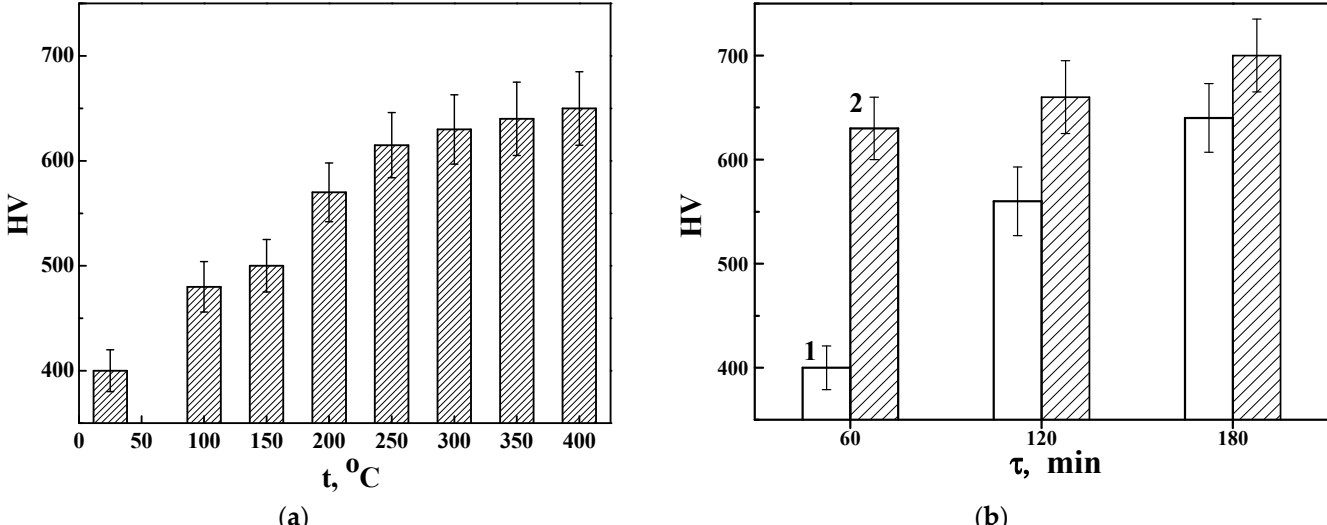

**Figure 11.** Changes in the microhardness HV of hard anodized layers on aluminum alloy 1011 depending on (**a**) heat treatment temperature t with specimens holding at it for 1 h and (**b**) duration of the anodizing process τ without heat treatment of specimens (1) and after holding them at 300 °C for 1 h (2).

The obtained effects of increasing the microhardness of specimens with anodized layers on the surface of aluminum alloy 1011 for different durations of their synthesis are associated with the dehydration of aluminum oxide. After all, the longer the duration of the synthesis of the anodized layer, the less water is retained in it. The layer becomes denser and therefore its hardness increases. However, the increase in microhardness after additional heat treatment of anodized specimens is due to phase amorphization in the synthesized layer. An increase in the relative amount of the $\alpha$-$Al_2O_{3(amorphous)}$ phase, which is characterized by high hardness, and a decrease in the content of the $\gamma$-$Al_2O_{3(amorphous)}$ phase, which has a slightly lower hardness, explain all the effects of changes in hardness after heat treatment.

### 3.4. Negative Manifestations of Heat Treatment on Hard Anodized Layers

Despite the increase in hardness of hard anodized layers after heat treatment, a network of microcracks appeared on their surface. Typical concentric and radially oriented cracks relative to the centre of a disk specimen (40 mm in diameter and 5 mm thick) are shown in Figure 12. Their appearance was regarded as a negative effect of heat treatment of specimens with an anodized layer. Such segmentation of the anodized layer by cracks was not found on the surface of the specimens heat-treated at temperatures below 300 °C.

The results of the phase analysis of specimens without and after heat treatment, presented above in Figures 8 and 9, convincingly confirmed the dehydration of the anodized layer after heating, causing a decrease in its volume. The base of the 1011 aluminum alloy specimen resists surface layer compression, resulting in tensile residual stresses in it. As a result, the critical level of residual tangential tensile stresses was reached in the surface layer of anodized specimens heated to 300 °C. Additionally, this was sufficient for the occurrence of cracks, radially oriented relative to the center of disk-shaped specimens. Circular cracks appeared within the areas fragmented by radial cracks, moving away from the centre of disk specimens. Their occurrence is also associated with residual tensile stresses, but acting in the radial direction. In addition, the 1011 alloy specimens and the anodized layer on their surface had different thermal expansion coefficients. Consequently, their different ability to expand during heating and subsequent cooling of the specimens also contributed to the formation of residual stresses and, accordingly, a network of cracks on their surface. However, in any case, the presence of such a network of cracks on the surface of hard-anodized layers after their heat treatment in this mode is undesirable. After

all, such cracks in the anodized layers will definitely worsen their serviceability (especially in corrosive-hydrogenated environments).

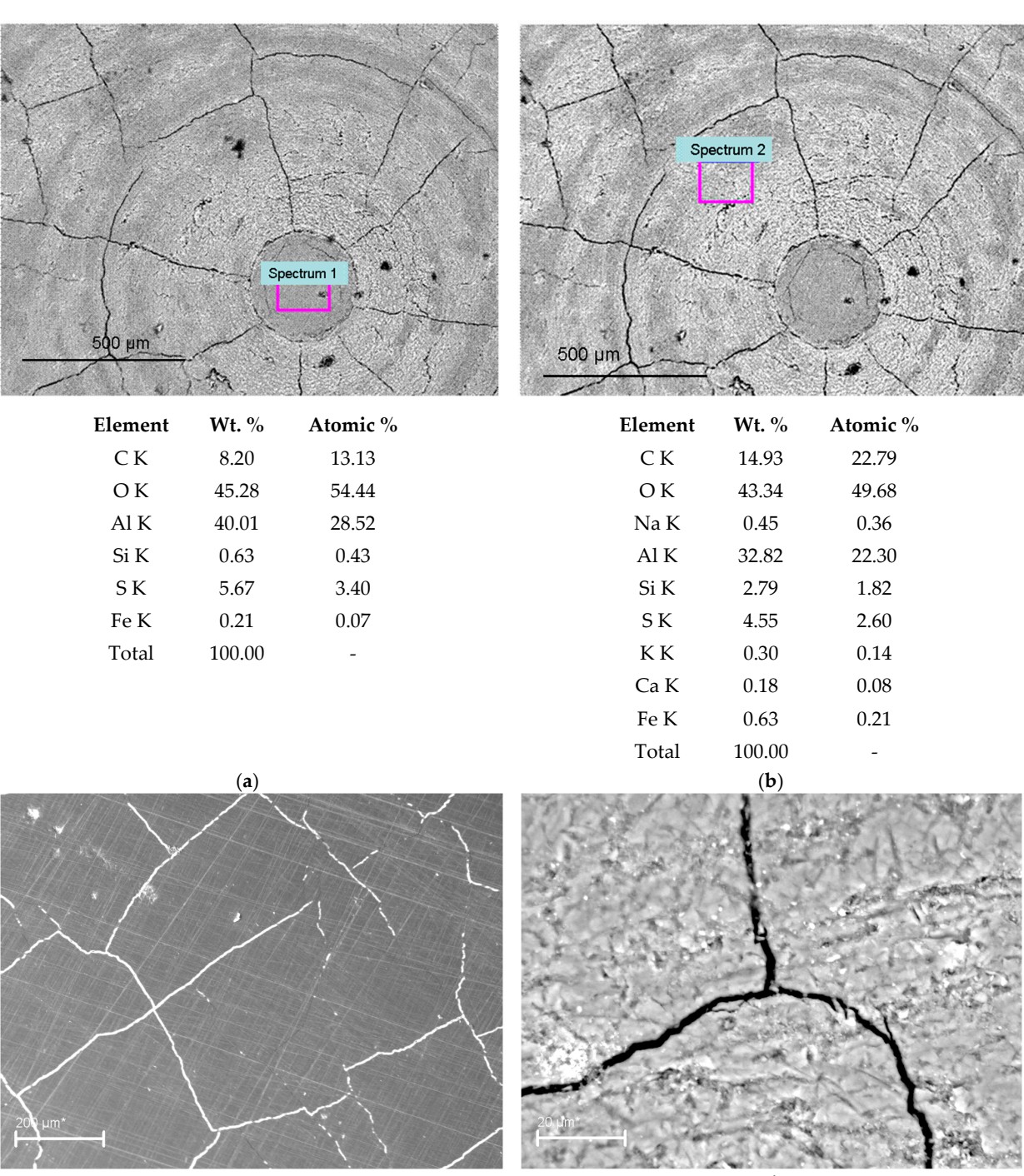

| Element | Wt. % | Atomic % |
|---------|-------|----------|
| C K | 8.20 | 13.13 |
| O K | 45.28 | 54.44 |
| Al K | 40.01 | 28.52 |
| Si K | 0.63 | 0.43 |
| S K | 5.67 | 3.40 |
| Fe K | 0.21 | 0.07 |
| Total | 100.00 | - |

| Element | Wt. % | Atomic % |
|---------|-------|----------|
| C K | 14.93 | 22.79 |
| O K | 43.34 | 49.68 |
| Na K | 0.45 | 0.36 |
| Al K | 32.82 | 22.30 |
| Si K | 2.79 | 1.82 |
| S K | 4.55 | 2.60 |
| K K | 0.30 | 0.14 |
| Ca K | 0.18 | 0.08 |
| Fe K | 0.63 | 0.21 |
| Total | 100.00 | - |

**Figure 12.** (**a**,**b**) Morphological features of the network of cracks in the hard anodized layer from the side of its outer surface with the results of micro-X-ray spectral analysis over the area different parts of the layer surface and (**c**,**d**) details of cracking at a higher resolution. The anodized layer was synthesized at −4–0 °C for 2 h, followed by heat treatment at 300 °C for 1 h.

In addition, one cannot exclude the effect on cracking of hard anodized layers of such a factor as stress concentration near sufficiently large particles of intermetallic compounds in their structure, shown in Figure 6. After all, the cracking of the anodized layer can be facilitated in areas with their high density. According to the results of micro-X-ray

spectral analysis of areas with cracks, shown in Figure 12b, an increased content of iron was recorded here. Namely, iron is one of the components of intermetallic compounds in the original structure of aluminum 1011, which are retained in the structure of the anodized layer.

### 3.5. Abrasive Wear Resistance of 1011 Aluminum Alloy with Hard Anodized Layers on Its Surface

The results presented in Figure 13 showed that the abrasive wear resistance of specimens with hard anodized layers increased both with an increase in the duration of the synthesis process and with the use additional heat treatment. In particular, the abrasive wear resistance of 1/W of specimens with anodized layers on the 1011 alloy increased even without additional heat treatment with increasing synthesis time. However, the positive effect of the duration of anodizing on the abrasive wear resistance of the specimens was practically leveled at a synthesis time of more than 2 h.

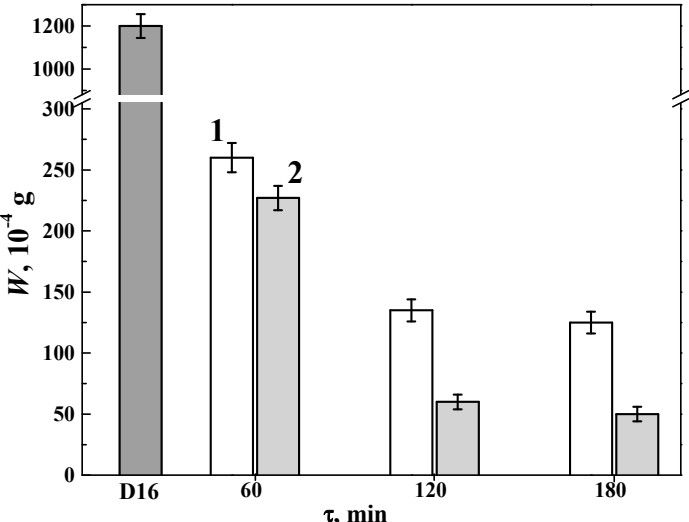

**Figure 13.** Effect of the duration $\tau$ of the synthesis of hard anodized layers on aluminum alloy 1011 on the mass loss W of specimens not subjected to heat treatment after synthesis (**1**) and after additional heat treatment at 300 °C for 1 h (**2**). Data for aluminum alloy D16 (2024 ISO) are given as a benchmark for comparing anodized layers on alloy 1011 in terms of their abrasive wear resistance 1/W.

As follows from Figure 13, heat treatment of specimens with anodized layers for 1 h at 300 °C additionally increased their abrasive wear resistance, regardless of the duration of the synthesis of these layers. In particular, the wear resistance of specimens anodized for 1, 2 and 3 h increased by 15%, 125% and 150%, respectively. In addition, as can be seen from Figure 13, the abrasive wear resistance of the specimens anodized for 2 and 3 h increased significantly (by 3.7 and 4.5 times, respectively) after their heat treatment at 300 °C compared with the corresponding value of 1/W for the layer synthesized for 1 h. Hence, it is obvious that the abrasive wear resistance of specimens with anodized layers after their heat treatment increased even more compared with specimens not subjected to heat treatment. Therefore, shown in Figure 12, the cracking of the anodized layer after heat treatment at a temperature of 300 °C did not change the generally positive effect of such treatment on the abrasive wear resistance of the specimen. Although it is quite possible that the detected cracking of the anodized layer could, to a certain extent, reduce the positive effect of heat treatment on the abrasive wear resistance of the anodized specimens.

To certify the modes of synthesis of anodized layers on the surface of specimens from alloy 1011 and their additional heat treatment, the values of abrasive wear resistance obtained for them were compared with the values of 1/W for alloy D16 (as one of the hardest and most common aluminum alloys). A significant positive effect of hard anodizing on the abrasive wear resistance of the 1011 alloy compared with the D16 alloy is shown

in Figure 13. Moreover, this effect increased both with an increase in the duration of the layer synthesis, and additionally with the use of heat treatment. The maximum increase in abrasive wear resistance (24 times) was achieved after the synthesis of the anodized layer for 3 h with heat treatment at 300 °C for 1 h. Additionally, its minimum increment (4.6 times) was obtained on specimens anodized for 1 h without subsequent heat treatment. It is clear that such a significant increase in the abrasive wear resistance of specimens with hard anodized layers opens up the prospect of a wider application of this method for surface hardening of aluminum alloys. This is especially important for elements operating in difficult conditions, when it is technologically impossible to exclude the influence of abrasive particles.

Photographs of characteristic traces of friction on the surface of specimens with the hard anodized layers, tested for abrasive wear resistance under conditions of a rigidly fixed abrasive, are shown in Figure 14a,b. Both specimens were anodized for 2 h. The first of them was not subjected, and the second was subjected to heat treatment at 300 °C for 1 h after the completion of the synthesis process. The results of a X-ray microspectral analysis over the area of these specimens are presented in Figure 14c,d. It can be seen that the composition of the anodized layer remained practically unchanged after additional heat treatment of the anodized specimen.

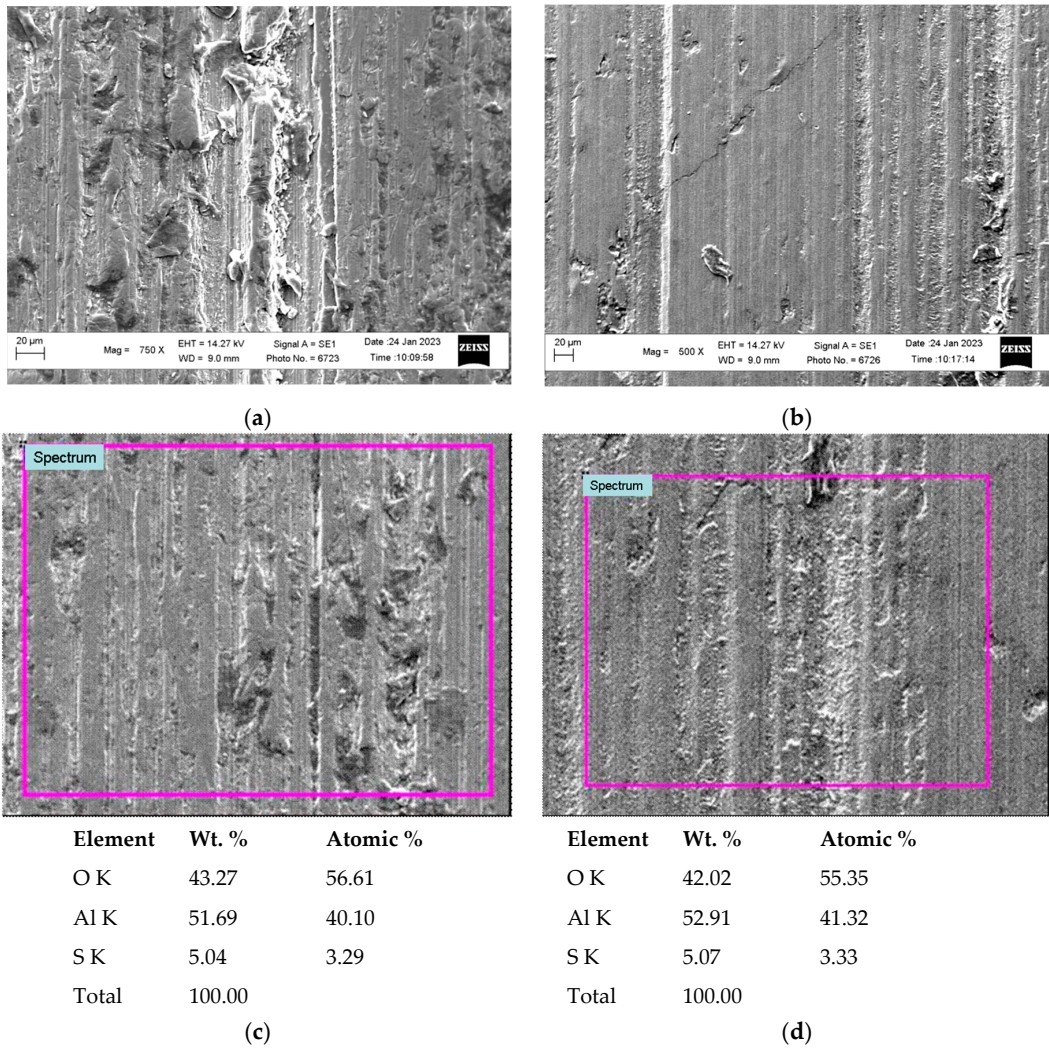

| Element | Wt. % | Atomic % | | Element | Wt. % | Atomic % |
|---------|-------|----------|---|---------|-------|----------|
| O K | 43.27 | 56.61 | | O K | 42.02 | 55.35 |
| Al K | 51.69 | 40.10 | | Al K | 52.91 | 41.32 |
| S K | 5.04 | 3.29 | | S K | 5.07 | 3.33 |
| Total | 100.00 | | | Total | 100.00 | |

(c)                             (d)

**Figure 14.** (**a**,**b**) Characteristic traces of friction during tests for abrasive wear resistance and (**c**,**d**) results of X-ray microspectral analysis of the area of damaged surfaces of specimens with layers anodized for 1 h, (**a**,**c**) obtained without heat treatment and (**b**,**c**) after heat treatment at 300 °C for 1 h.

Damage to the surfaces of these specimens occurred through the mechanism of local cutting of the anodized layer by abrasive particles from the counterbody (from a corundum disk), which performed the function of microcutters. However, the distribution over the area and the depth of friction marks in the form of grooves on the surfaces of both specimens differed significantly. So, on the surface of a specimen with a lower hardness, not subjected to heat treatment, the depth and number of grooves left by abrasive particles were greater, and the distance between them was less than on the surface of heat-treated HAL. In addition, in the damaged areas of Figure 14a,c, a large number of zones with signs of local adhesion of friction elements can be observed. All these features of the damaged anodized layer are consistent with the lower surface hardness of this specimen. In the specimen subjected to heat treatment, the contact traces of corundum particles with the anodized surface were wider and less deep than in the specimen without heat treatment. As shown in Figure 14b,d, the traces of contact of corundum particles with an anodized surface on the specimen subjected to heat treatment were wider and less deep than in the specimen without heat treatment. At the same time, there were practically no signs of local adhesion between the friction elements. These signs of damage are quite consistent with higher hardness and abrasive wear resistance of the HAL after heat treatment.

## 4. Discussion

At the present stage of development of surface hardening technologies, several of them are used to improve the abrasive wear resistance of aluminum alloys. Among them are not only the synthesis of hard anodized layers (HAL), but also plasma-electrolyte oxidation (PEO) of surfaces and methods of thermal spraying of coatings, in particular, such as spraying of vanadium carbide powder using high-velocity oxygen fuel (HVOF) and arc-sprayed coating (ASC) by cored wire. In the Table 1 shows the main properties that determine the operational characteristics of surface-hardened layers on aluminum alloys obtained by each of these methods. D16 aluminum alloy was used as the abrasive wear resistance standard, and data for high-strength steel 100Cr6 were also given for comparison. It is obvious that the synthesis of hard anodized layers today provides worse wear resistance of aluminum alloys than other analyzed methods. However, this method is highly productive, relatively cheap, environmentally safe, and, most importantly, does not require finishing machining (unlike the other methods listed above) and is already widely used in many engineering industries. Therefore, scientific research aimed at further increasing the abrasive wear resistance of aluminum alloys by the method of synthesis of hard anodized layers on their surface is very important from a practical point of view. Moreover, even small progress in increasing the wear resistance of HAL is positively appreciated by industrial users. The HVOF and ASC methods are commonly used for wear protection not only for new parts, but also for the restoration of wear-damaged parts. PEO and hard anodizing techniques are mainly used to protect and prevent wear on the surfaces of new parts.

In conclusion, it should be noted that the use of the method of synthesis of hard anodized layers increases the abrasive wear resistance of parts made of aluminum alloys to the level characteristic of steel elements. This creates the prospect of replacing steel and cast-iron machine elements with aluminum ones with hardened surface layers, which will significantly reduce the weight of both individual elements and the entire structure as a whole. In addition, this replacement helps to reduce carbon dioxide emissions into the atmosphere. After all, the technological processes of smelting steels and aluminum alloys differ significantly in the amount of harmful emissions into the atmosphere, with the advantage of aluminum alloys.

**Table 1.** Ranking of the surface layers of aluminum alloys, hardened in different ways, according to the properties that determine their functional ability.

| Spraying Methods | Relative Abrasive Wear Resistance * of Specimens Obtained During Tests | | Microhardness at Loading 50 g, HV | Relative Energy Costs ** for Forming a Hardened Layer with a Thickness of 100 Microns on an Area of 1 m$^2$ | Ecological Advantages and Disadvantages of the Analyzed Processes |
| --- | --- | --- | --- | --- | --- |
| | with a Rigidly Fixed Abrasive | with Non-Fixed Abrasive | | | |
| Standart for comparison: aluminium alloy D16 (2024 ISO) | 1 | 1 | 110 | — | — |
| 100Cr6 steel (SAE 52100) (HRC 64) | 20 | 3 | 800 | — | $CO_2$ emissions to the atmosphere |
| Galvanic chrome plating | 35 | 5 | 1000 | 3 | Carcinogenic electrolytes |
| HVOF spraying by VC carbides | 75–85 | 7–10 | 1100, with carbide VC microhardness to 2500 HV | 7 | Noise level: 130 dB; dust from micron-sized particles |
| PEO | 70–90 | 8–10 | 1900 | 7 | Eco-friendly electrolytes |
| HAL | 20 | 3 | 700 | 2 | Eco-friendly electrolytes |
| ASC | 30 | 5 | 750, with oxides microhardness to 800–2000 HV | 1 | Noise level: 120 dB; dust from micron-sized particles |

*—the relative change in the resistance to abrasive wear of hardened layers formed by any of the analyzed methods was evaluated relative to the corresponding properties of the D16 alloy, taken as a standard for comparison; **—the relative change in energy consumption for the formation of a hardened surface layer by any of the analyzed methods was estimated relative to the corresponding costs for the formation of an ASC from cored wires with a thickness of 100 μm on a surface of 1 m$^2$, taken as a standard for comparison.

In conclusion, it should be noted that the use of the method of synthesis of hard anodized layers increases the abrasive wear resistance of parts made of aluminum alloys to the level characteristic of steel elements. This creates the prospect of replacing steel and cast-iron machine elements with aluminum ones with hardened surface layers, which will significantly reduce the weight of both individual elements and the entire structure as a whole. In addition, this replacement helps to reduce carbon dioxide emissions into the atmosphere. After all, the technological processes of smelting steels and aluminum alloys differ significantly in the amount of harmful emissions into the atmosphere, with the advantage of aluminum alloys.

## 5. Conclusions

It was shown metallographically that large particles of intermetallides in the 1011 aluminum alloy slow down the advance of the anodization front, bend it, and remain inside the hard anodized layer. Under the influence of electrolyte, micron-sized pores are formed in their place, which worsen the functional properties of hard-anodized layers.

Dehydration of hard-anodized layers with an increase in the duration of their synthesis was confirmed by X-ray phase analysis. Thus, after 1 h of anodizing, gibbsite $Al_2O_3$ $3H_2O$ with three water molecules predominated in the layer structure, and after 3 h only one water molecule remained in aluminum oxide.

Aluminum oxides $\alpha$-$Al_2O_{3(amorphous)}$ and $\gamma$-$Al_2O_{3(amorphous)}$ were found in the synthesized hard anodized layers after their heat treatment at 300 °C for 1 h. Their appearance was considered a sign of further dehydration of aluminum oxide.

Due to the dehydration of hard anodized layers on aluminum alloy 1011, their microhardness HV increased both with an increase in the duration of synthesis and after additional heat treatment. The maximum microhardness (700 HV) was achieved by the combined action of both of these factors.

Cracking of hard anodized layers after their heat treatment at temperatures starting from 300 °C was revealed. The appearance of cracks was explained by residual stresses

arising from the dehydration of the hard anodized layers and the discrepancy between the thermal expansion coefficients of these layers and the aluminum base. Cracking of anodized layers reduces the possibility of their use after heat treatment above 200 °C (especially when operating in aggressive environments).

A significant increase in abrasive wear resistance (with a fixed abrasive) of specimens with hard anodized layers on the surface of 1011 aluminum alloy was shown both after an increase in the duration of the synthesis process and after subsequent heat treatment. The maximum (compared to alloy D16) increase in abrasive wear resistance (24 times) was achieved after anodizing the specimens for 3 h with heat treatment at 300 °C for 1 h.

The ranking of various methods of hardening aluminum alloys showed that the abrasive wear resistance of HAL is 20 times higher in comparison with the D16 alloy. At the same time, plasma-electrolyte oxidation increased the resistance of the D16 alloy to abrasive wear by 70–90 times, and high-velocity oxygen fuel spraying of vanadium carbide powder by 75–85 times. However, both of these methods are complex and energy-intensive, changing the dimensions of parts and requiring their fine grinding. Additionally, although HAL have less wear resistance, nevertheless, their synthesis is much cheaper and no fine-tuning of parts is required. Moreover, HAL are already used in industry to harden engine pistons, clamshell rotators, and pulleys. Therefore, the obtained improvement in the properties of HAL by their heat treatment makes it possible to increase the service life of already industrially processed parts and further expand the directions of their application.

**Author Contributions:** Conceptualization, M.S. and I.P.; methodology, V.H. and K.Z.; validation, I.K., V.H. and K.Z.; formal analysis, I.P., J.P. and H.C.; investigation, V.H., I.K. and K.Z.; Interpretation of results, M.S., I.P., O.S., J.P., V.H. and H.C.; resources, J.P. and I.P.; data curation, I.P., M.S., V.H. and J.P.; writing—original draft preparation, M.S., V.H., H.C. and O.S.; discussion, M.S., I.P., O.S., J.P., H.C. and V.H.; writing—review and editing, M.S., O.S., J.P. and H.C.; visualization, V.H., I.K. and K.Z.; supervision, I.P. and M.S.; project administration, J.P. and H.C.; funding acquisition, J.P. All authors have read and agreed to the published version of the manuscript.

**Funding:** The results of the research presented in the article were obtained within the framework of the project "Development of methods for the synthesis of hard anodic wear-resistant layers on structural aluminum alloys", 0120U100779; Support for priority state scientific research and scientific and technical (experimental) developments of the Department of Physical and Technical Problems of Materials Science of the National Academy of Sciences of Ukraine on the basis Resolution of the Presidium of the National Academy of Sciences of Ukraine dated 18.12.2019 No. 339 "On Approving the Allocation of Budgetary Funding of the National Academy of Sciences of Ukraine for 2020" based on the Resolution of the Presidium of the National Academy of Sciences of Ukraine dated 23.12.2020 No. 296 "On Budgetary Funding of the National Academy of Sciences of Ukraine in 2021" (Karpenko Physico-Mechanical Institute of the NAS of Ukraine, 2020-2021, 18, state budget funds).

**Institutional Review Board Statement:** Not applicable.

**Informed Consent Statement:** Not applicable.

**Data Availability Statement:** The data presented in this study are available upon request from the corresponding author.

**Conflicts of Interest:** The authors declare no conflict of interest.

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
