# Peer review of "The Effect of Heat Treatment on the Structural-Phase State and Abrasive Wear Resistance of a Hard-Anodized Layer on Aluminum Alloy 1011"

_coatings, doi:10.3390/coatings13020391_

Round 1

Reviewer 1 Report

I have had the opportunity to review the article " The effect of heat treatment on the structural-phase state and abrasive wear resistance of a hard-anodized layer on aluminum alloy 1011". In order to get the full potential of the paper, some corrections need to be done. Some comments are provided below:

Abstract: writing is too generalized. The main theme of this paper is not described in the abstract. Abstract section should be concisely reflected the content and summarize the problem, the method, the results, and the conclusions. The abstract needs to be improved. Demonstrate in the abstract novelty, practical significance. Also, please add more qualitative and quantitative results of your work.

In the introduction section, more literature paper have to be included to explain the subject better. The introduction section has been written beautifully but need to include recent published papers regarding your work (Please refer to most relevant papers).

Some giving citations need to be check such as [5-9], [25-28], as they may not provide the required information in a sentence. Each one of the cited references  must be discussed individually and demonstrate their significance to your work. Besides, add scientific novelty and practical relevance. Add a clear purpose to the article. Please show the literature gaps demonstrating the presented study fills it. At the last paragraph of the introduction, please clearly show the general outline of the paper and show the importance of the study along with the main aim.

References are not enough. Such a work deserves many citations related with the machinability, optimization and finite element methods. Minimum 10-15 references need to be added and some of them should be discussed.

The authors can give the actual photos belong to experimental infrastructure. Also, a graphical abstract would be helpful for the presentation of the study. (preferable)

How were the process parameters selected?

Indeed, there are an impressive amount of results. However, the conclusions section needs to improve with selected and highlighted main findings. In conclusion section, it is necessary to more clearly show the novelty of the article and the advantages of the proposed method. Add qualitative and quantitative results of your work. Please try to emphasize your novelty, put some quantifications, and comment on the limitations. This is a very common way to write conclusions for a learned academic journal. The conclusions should highlight the novelty and advance in understanding presented in the work. This is a very common way to write conclusions for a learned academic journal.

Language used in the manuscript is generally satisfying. However, writers should pay more attention of singular / plural nouns. Also, they should control the spell check/ punctuation of words and sentences. Please check all manuscript for language and misspellings. Also, please recheck upper and lower case letter. In addition, spaces should be added between words and numbers. The authors can use suitable grammar-checking software / use the help of a native English speaker to correct these mistakes. Please fix the typographical and eventual language problems in paper. Please recheck author names in the introduction some o

The alignment of some figures is wrong. Please revise it. Also please improve presentation of figures since their current form is very simple, and title of axis are inconsistencies with each other some of them bigger and some of them lower etc.

Author Response

Manuscript ID: coatings-2165463
Type of manuscript: Article
Title: The effect of heat treatment on the structural-phase state and
abrasive wear resistance of a hard-anodized layer on aluminum alloy 1011
Authors: Mykhailo Student *, Iryna Pohrelyuk *, Juozas Padgurskas, Volodymyr
Hvozdets’kyi, Khrystyna Zadorozna, Halyna Chumalo, Oleksandra Student, Ihor
Kovalchuk
Received: 30 December 2022

The authors of the article are grateful to reviewer for the in-depth analysis of the results presented in our article. In our opinion, after the additions made to the text of the article and corrections of inaccuracies made by us the article became more understandable to specialists searching the new methods increasing the key functional properties of aluminium alloys for expand the limits their application in different branches of techniques where the advantages of these alloys are important (machine and aircraft engineering, space vehicles and military equipments, etc.).

The following changes were made to the article in responds to reviewer’s remarks.

Reviewer N1

  1. The authors agree with the reviewer's remark according the abstract and rewrote its taking into account made recommendations.

  1. The authors agree with the reviewer's remark regarding the "Introduction" section. In order to take into account his recommendations and better explain the subject of our study, this section has been supplemented with text analyzing current articles and expanding the list of links to recent publications.

  1. In response to the reviewer's comments, the authors carefully checked all the listed references for their correspondence with the text and replaced two of them (27 and 28) with more appropriate ones. All other links fully correspond to their description in the text of the article.

Moreover, in the additionally provided text to the "Introduction" section with an analysis of literary sources, the authors tried as much as possible within the given time frame to demonstrate the gaps in the research topic and substantiate the purpose of our study aimed at filling these gaps.

  1. The authors partially took into account the comment of the reviewer and expanded the list of references to literary sources on the topic of the study and added references from 34 to 51. Only references related to the use of the finite element method for modeling the hard anodizing process were not included in this list, since this is a topic for a separate study.

  1. The authors added a photograph of the setup and expanded the description of the methods the abrasive wear resistance determineation in conditions of fixed and non-fixed abrasive, and also detailed the requirements that were followed during the tests.

  1. In our study, we used the widely tested and most common electrolyte and synthesis modes used both in research and on an industrial scale to form hard-anodized layers [https://waykenrm.com/blogs/hard-coat-anodizing-of-aluminum/, https://aerospacemetalsllc.com/what-is-hardcoat-anodizing/, https://www.anodizeusa.com/anodizing-systems-hard-coat.php].

  1. The conclusions in the article were maximally revised with taking into account of reviewer’s remarks.

  1. The reviewer's comments were taken into account and appropriate corrections were made to the text of the article. The entire text of the article has been checked for possible errors listed by the reviewer.

  1. The reviewer's comments were taken into account and the necessary corrections were made to the figures.

Reviewer 2 Report

The manuscript has a thorough investigation and is therefore worthy of publication. However, the following issues have to be addressed before the final acceptance;

1) The introduction should be more focused. The effects of HAL on wear and friction should be discussed in detail. Perspectives of industrial applications should also be mentioned and discussed.

2) In line 131, it is written, ‘Nanosized Al2Cu inclusions’. How does the nanosize determined? Experimental evidence is needed.

3) In line 135, it is written, ‘significant amount of sulfur’. The significance of the S in wear reduction should be mentioned and discussed.

4) The SEM of wear tracks after the wear tests are required to understand the wear mechanisms. Compositional analyses are required to understand the abrasive particles. Any evidence of HAL delamination?

5) The SEM and EDS analyses of counterfaces are also required. The role of transfer materials from the disc may have a role in wear processes.

6) How does the wear test parameters were determined? Are they optimized?

Author Response

Manuscript ID: coatings-2165463
Type of manuscript: Article
Title: The effect of heat treatment on the structural-phase state and
abrasive wear resistance of a hard-anodized layer on aluminum alloy 1011
Authors: Mykhailo Student *, Iryna Pohrelyuk *, Juozas Padgurskas, Volodymyr
Hvozdets’kyi, Khrystyna Zadorozna, Halyna Chumalo, Oleksandra Student, Ihor
Kovalchuk
Received: 30 December 2022

The authors of the article are grateful to reviewer for the in-depth analysis of the results presented in our article. In our opinion, after the additions made to the text of the article and corrections of inaccuracies made by us the article became more understandable to specialists searching the new methods increasing the key functional properties of aluminium alloys for expand the limits their application in different branches of techniques where the advantages of these alloys are important (machine and aircraft engineering, space vehicles and military equipments, etc.).

The following changes were made to the article in responds to reviewer’s remarks.

Reviewer N2

  1. The authors took into account the comment of the reviewer and expanded the "Introduction" section. At the same time, attention was given to the problem of HAL wear resistance and the possibility of expanding their use in industry.

  1. The reviewer is right. Unfortunately, we have no direct experimental confirmation of the presence of Al2Cu nanosized inclusions in the structure of the anodized layer. Such assumption was made on the basis of indirect results. Namely, a certain amount of copper was recorded in the anodized layer by its X-ray spectral analysis over the area. At the same time, copper was not found in the composition of micron sized Fe(Mn,Cu, Mg)3Al intermetallic compounds. On this basis, the assumption was made that copper can be present in the structure of the anodized layer in the form of nanosized Al2Cu intermetallic compounds. Therefore, we would like such an explanation to remain in the text of the article.

  1. This remark of the reviewer was not taken into account by the authors. Really anodized layers contained up to 6.5 wt. % sulfur. This is a consequence that it is impossible to exclude the permeation of sulfur into the anodized layer synthesized in a sulfate electrolyte. Therefore, it is not possible to compare the anodized layers with and without sulfur in its composition to establish its role in wear. It is possible, of course, to synthesize an anodized layer without the presence of sulfur in it, when the synthesis was carried out in oxalic or citric acid, but then the comparison in terms of wear resistance will be incorrect.

  1. The authors took into account the comment of the reviewer. Photographs of characteristic friction marks on the surface of specimens with a hard anodized layer, tested for abrasive wear resistance under conditions of a rigidly fixed abrasive, have been added to the text of the article (Fig. 14). One of the specimens was subjected to heat treatment, while the second was not. The features of damage to the surfaces of each of them were also described.

  1. This remark of the reviewer is not accepted. An electron microscope study of the friction surface of the abrasive disk was not carried out, since it was necessary to destroy it in order to place it in the electron microscope chamber. Therefore, we do not have such data.

  1. We have the following comments to this reviewer's remark. The properties of coatings with high microhardness (up to 2000 HV) are usually investigated in our laboratory. Therefore, an abrasive disc made of electrocorundum was chosen for research, the microhardness of grains of which reaches 2200 HV. These abrasive discs from the same purchase batch have been used in our laboratory for 20 years. This allows us to compare the results of 20 years ago with modern ones. The load of pressing the disc against the specimen was chosen so that after 30 minutes of testing it was possible to fix the weight loss of both very hard coatings (PEO, HVOF) and aluminum base with low hardness. This made it possible to clearly record the weight loss of both coatings and materials used as standards.

Author Response

Manuscript ID: coatings-2165463
Type of manuscript: Article
Title: The effect of heat treatment on the structural-phase state and
abrasive wear resistance of a hard-anodized layer on aluminum alloy 1011
Authors: Mykhailo Student *, Iryna Pohrelyuk *, Juozas Padgurskas, Volodymyr
Hvozdets’kyi, Khrystyna Zadorozna, Halyna Chumalo, Oleksandra Student, Ihor
Kovalchuk
Received: 30 December 2022

The authors of the article are grateful to reviewer for the in-depth analysis of the results presented in our article. In our opinion, after the additions made to the text of the article and corrections of inaccuracies made by us the article became more understandable to specialists searching the new methods increasing the key functional properties of aluminium alloys for expand the limits their application in different branches of techniques where the advantages of these alloys are important (machine and aircraft engineering, space vehicles and military equipments, etc.).

The following changes were made to the article in responds to reviewer’s remarks.

Reviewer N3

  1. Reviewer's comment taken into account. The keywords are simplified and presented in a more logical sequence.

  1. Reviewer's comment taken into account. In the "Introduction" section, text was added to the article, which more clearly substantiated the choice of the hard anodizing method and the perspective of its heat treatment to increase the hardness and abrasive wear resistance of the anodized layers.

  1. Reviewer's comment taken into account. A description of the equipment used for heat treatment and its parameters has been added to the article in the "Materials and Methods" section.

  1. The authors are grateful to the reviewer of the article for this remark and corrected the errors in the numbering of headings in the second section of the article.
  2. The authors took into account the comment of the reviewer and replaced in the text of the article all the expressions "resistance to abrasive wear" with "abrasive wear resistance".
  3. The authors do not agree with the opinion of the reviewer and insist on the correct location of this part of the text of the article in the "Discussion" section. After all, this section discusses not literary, but data obtained by the authors. They were used to rank the effect of hard anodizing with additional heat treatment on the corresponding properties of surface hardened layers on 1011 aluminum alloy. In support of this statement, a description of surface hardening methods, which are used in Table 1, has been added to the methodological part of the article. Their properties were used for comparison with the properties of the surface layer of 1011 aluminum alloy after its hard anodization and subsequent heat treatment. Therefore, in our opinion, these data should be placed in the "Discussion" section.
  4. We partially took into account the reviewer’s comments. In formulating the conclusions, we proceeded from the fact that readers usually make a decision about the need to read the entire article based on them. However, we still tried to take into account this remark of the reviewer. Therefore, the conclusions in the article were maximally (taking into account our position) reduced.

Round 2

Reviewer 3 Report

It can be accepted now.